# Development of an IoT-Based Device for Data Collection on Sheep and Goat Herding in Silvopastoral Systems

**DOI:** 10.3390/s24175528

**Published:** 2024-08-26

**Authors:** Mateus Araujo, Paulo Leitão, Marina Castro, José Castro, Miguel Bernuy

**Affiliations:** 1Research Center in Digitalization and Intelligent Robotics (CeDRI), Instituto Politécnico de Bragança, Campus de Santa Apolónia, 5300-253 Bragança, Portugal; pleitao@ipb.pt; 2Laboratório Associado para a Sustentabilidade e Tecnologia em Regiões de Montanha (SusTEC), Instituto Politécnico de Bragança, Campus de Santa Apolónia, 5300-253 Bragança, Portugal; marina.castro@ipb.pt; 3Centro de Investigação de Montanha (CIMO), Instituto Politécnico de Bragança, Campus de Santa Apolónia, 5300-253 Bragança, Portugal; 4Instituto Politécnico de Bragança, Campus de Santa Apolónia, 5300-253 Bragança, Portugal; mzecast@ipb.pt; 5Universidade Tecnológica Federal do Paraná, Campus Cornélio Procópio, Cornélio Procópio 86300-000, Brazil; angel@utfpr.edu.br

**Keywords:** IoT, GNSS, grazing activity, sheep and goats, LoRa, silvopastoral systems, data collection

## Abstract

To evaluate the ecosystem services of silvopastoral systems through grazing activities, an advanced Internet of Things (IoT) framework is introduced for capturing extensive data on the spatial dynamics of sheep and goat grazing. The methodology employed an innovative IoT system, integrating a Global Navigation Satellite System (GNSS) tracker and environmental sensors mounted on the animals to accurately monitor the extent, intensity, and frequency of grazing. The experimental results demonstrated the high performance and robustness of the IoT system, with minimal data loss and significant battery efficiency, validating its suitability for long-term field evaluations. Long Range (LoRa) technology ensured consistent communication over long distances, covering the entire grazing zone and a range of 6 km in open areas. The superior battery performance, enhanced by a solar panel, allowed uninterrupted operation for up to 37 days with 5-min interval acquisitions. The GNSS module provided high-resolution data on movement patterns, with an accuracy of up to 10 m after firmware adjustments. The two-part division of the device ensured it did not rotate on the animals’ necks. The system demonstrated adaptability and resilience in various terrains and animal conditions, confirming the viability of IoT-based systems for pasture monitoring and highlighting their potential to improve silvopastoral management, promoting sustainable practices and conservation strategies. This work uniquely focuses on documenting the shepherd’s role in the ecosystem, providing a low-cost solution that distinguishes itself from commercial alternatives aimed primarily at real-time flock tracking.

## 1. Introduction

Silvopastoral systems have gained increasing attention in mountain and marginal areas, as livestock are recognized as an essential component of healthy landscapes [1,2,3]. This focus is based on the understanding that properly managed grazing can have a positive impact on vegetation structure and nutrient recycling, and help to conserve biodiversity [4,5,6,7]. However, a detailed understanding of the ecological role of grazing—shaped by its extent, intensity, and frequency—is a challenging task. These factors, which are crucial for assessing grazing’s ecosystem services, are difficult to quantify precisely due to the inherently dynamic nature of animal movements and the myriad of interactions within ecosystems. This complexity has led to a pressing demand for innovative data collection and ecological analysis technologies [8,9,10,11,12,13,14,15,16,17].

The advent of IoT technologies has marked a new era in spatiotemporal biological monitoring, offering the potential to transform ecological research by providing detailed, real-time, and continuous data collection capabilities [18,19]. IoT’s potential to offer unprecedented insights into livestock grazing patterns and their subsequent environmental impacts is immense [20,21]. However, despite IoT’s clear advantages to the field, its application in elucidating the ecological implications of grazing is still embryonic. Most existing studies primarily focus on the technological dimensions of IoT applications instead of their ecological insights, leaving a gap in the literature that still needs full addressing [22].

This dichotomy in research focus underscores a broader debate within the scientific community regarding the most effective methodologies for monitoring grazing activities. Traditional observational techniques revered for their directness and historical data continuity are juxtaposed against the precision and comprehensiveness of modern technological solutions like IoT [23,24,25]. Despite the potential of IoT to revolutionize ecological monitoring by enabling a finer granularity of data collection on animal movements and landscape interactions, questions regarding its reliability, potential biases, and the need for its validation against conventional methods remain pertinent concerns [17].

Studies on the impact of grazing on ecosystems often have contradictory results. Some researchers support controlled grazing to maintain biodiversity and soil health [26,27,28], while others highlight the risks of overgrazing, including soil erosion, reduced plant diversity and habitat degradation [29,30,31,32]. This divergence of opinions emphasizes the need for a nuanced, data-driven approach to grazing management, informed by accurate and timely monitoring of livestock activities.

The current study tackles these challenges by leveraging IoT technology to comprehensively monitor sheep and goat grazing patterns. It aims to fill the research voids identified by previous studies, offering a rich dataset on grazing dynamics to inform sustainable management practices. This research endeavor advances the technological frontiers of ecological monitoring and actively engages in the scientific debate regarding pastoralism’s ecological and landscape impacts, striving to harmonize livestock productivity with ecosystem conservation goals.

The promise of IoT technologies in ecological research lies in their ability to capture precise, real-time data on livestock movements and environmental conditions [33]. Sensor technology, for instance, can shed light on the microclimatic experiences of grazing animals, while GPS tracking elucidates their movement patterns and preferences [34,35,36,37,38]. Such detailed information is instrumental in understanding the direct and indirect effects of grazing on plant diversity, soil composition, and overall ecosystem functionality.

Moreover, integrating IoT-generated data with Geographic Information System (GIS) platforms facilitates the spatial analysis of grazing impacts on a landscape level [37]. This capability is crucial for identifying ecologically sensitive areas, predicting potential overgrazing hotspots, and crafting targeted conservation strategies. Applying machine learning algorithms to the extensive datasets generated by IoT devices can also reveal patterns and trends not immediately apparent, providing predictive insights into the ecological outcomes of various grazing regimes.

Nevertheless, challenges, including data management concerns, the accuracy and reliability of sensor data, and ethical considerations surrounding animal welfare and the environmental footprint of the technology itself, temper the enthusiasm for IoT’s capabilities. Despite these hurdles, the consensus among researchers is gradually shifting toward a more integrated and technologically informed approach to understanding the impacts of grazing on ecosystems.

As this field of study evolves, fostering interdisciplinary collaboration will be vital to unlocking the full potential of IoT in ecological monitoring. The ultimate objective is to responsibly employ IoT technologies to develop grazing practices that sustain human livelihoods and ecosystem resilience. By applying IoT to the examination of sheep and goat grazing, this study validates the utility of modern technologies in silvopastoral monitoring and emphasizes their significance in deepening our comprehension of the complex interactions within pastoral ecosystems.

The current study tackles these challenges by leveraging IoT technology to comprehensively monitor sheep and goat grazing patterns. It aims to fill the identified research gaps, offering a rich dataset on grazing dynamics to inform sustainable management practices. This research endeavor advances the technological frontiers of ecological monitoring and actively engages in the scientific debate regarding pastoralism’s ecological and landscape impacts, striving to harmonize livestock productivity with ecosystem conservation goals. In contrast to existing market solutions that focus on real-time flock location, this work uniquely aims to document the shepherd’s role in the ecosystem, providing a low-cost solution to fairly recognize and remunerate shepherds, thereby enhancing their productivity and income.

The paper is structured as follows: Section 2 provides an overview of related works, highlighting key characteristics of existing research in the field. Section 3 brings the objectives and requirements of the Device developed. Section 4 details the device and system architecture, delineating the IoT framework. Section 5 delves into the practical deployment of the system data transmission, visualization techniques, device implementation on animals, installation locations, software availability, and ethical considerations. Section 6 presents the study results, addressing data transmission reliability, battery efficiency, system accuracy, and experimental interpretation and Section 7 engages in a comprehensive discussion of the findings, comparing them with relevant literature and exploring implications for future research directions. Finally, Section 8 provides concluding remarks, summarizing the contributions and benefits of this type of research, and points out future work.

## 2. Related Work

As already seen in the introduction, small ruminant tracking is essential for efficient herd management, monitoring grazing patterns and ensuring animal welfare. This section analyzes existing commercial solutions and relevant academic work in this field, highlighting their characteristics, advantages and limitations.

To respond to the need to track small ruminants, companies such as Domodis (Navarra, Spain) [39], Pastoral (London, UK) [40], Digitanimal (Madrid, Spain) [41], Nofence (Batnfjordsøra, Norway) [42] and Traki (New York, NY, USA) [43] have developed commercial solutions. Domodis is suitable for farm animals, including horses, sheep, cows, goats and wild animals, although there are concerns about the ergonomics of the device for goats and sheep. Pastoral provides a back-mounted tracking device that incorporates a solar panel, but this poses potential risks such as entanglement in vegetation for goats. Despite offering low-cost entry points, starting at $95, Pastoral’s devices generally have a shorter battery life compared to Domodis, which starts at 397 € per unit, and both also require monthly subscriptions for use.

Digitanimal offers devices with prices starting at 189 €, designed to be worn on the side of the animal’s neck. This positioning aims to balance comfort and functionality, allowing effective tracking without significantly hindering the animal’s movements. Digitanimal also requires a monthly subscription to use. Nofence, on the other hand, does not publicly disclose its prices and subscription but presents an interesting ergonomic design in which the entire device is positioned underneath the small ruminant, potentially offering a different set of benefits and challenges compared to other solutions on the market. The company Traki offers the cheapest of the commercial devices, costing 44.16 €. It has a 10,000 mAh battery that guarantees operation for 2 to 12 months. However, this device has a monthly fee starting at 11.30 €/month and its data collection varies between 15 s and 1 min, depending on the subscription.

All of the solutions, with the exception of Nofence and Traki, feature a unique device without a counterweight, which means that it can rotate on the animal’s neck while grazing. In addition, all of them, with the exception of Pastoral and Traki, have devices that initially cost more than 160 €, which makes them unaffordable in many cases. All of these devices feature satellite communication, General Packet Radio Service (GPRS), or similar, meaning that there is no need for a local physical architecture with a gateway, for example.

In summary, Domodis and Pastoral offer versatile solutions, but present ergonomic challenges for goats and sheep. Digitanimal proposes a design that seeks to balance comfort and functionality and Nofence stands out for its innovative design, positioning the device under the animal. Traki is the cheapest commercial device with the longest battery life.

In addition to commercial devices, there is academic work that, as well as tracking animals, combines technologies in order to collect more grazing data and test new approaches. For example, ref. [44] has developed a low-cost prototype that uses LoRa, GPS and accelerometer technologies to track and detect the movement of small ruminants in real-time. However, the battery life does not reach two days and the data are acquired at an interval of 60 s, suggesting better energy management of the sensors and the need for a solar panel to help with feeding.

The need for effective livestock monitoring is intensified by the lack of direct supervision and the challenging conditions of grazing environments, especially in regions with difficult access and limited communication infrastructure. A recent study presents an IoT-based monitoring system that integrates Low Power Wide Area Networks (LPWAN) technology, cloud services, and virtualization to provide real-time livestock location monitoring [45]. This system employs a wearable device equipped with inertial sensors, a GPS receiver, and a LoRaWAN transceiver, achieving a satisfactory balance between performance, cost, and energy consumption. The co-design of hardware and firmware is optimized to minimize energy use, resulting in significantly extended battery life. The system was evaluated through pilot tests in northern Italy, analyzing aspects such as packet delivery ratio, energy consumption, localization accuracy, battery discharge measurement, and delay.

To address the challenges of monitoring extensive livestock in hard-to-reach areas, another study presents a solution based on an LPWAN that operates both with and without internet connectivity [46]. The proposed system includes a wearable device with inertial sensors, GPS, and wireless communication, combined with LPWAN infrastructure that supports real-time monitoring and logging of cattle positions and activities. The hardware and firmware design achieve extremely low energy consumption, allowing for months of battery life. The system has been rigorously tested in various laboratory setups and evaluated in real-world scenarios in mountainous and forested areas, demonstrating its effectiveness in extreme environments and its ability to provide detailed and reliable monitoring data for livestock management.

As an example of integration, ref. [47] used GNSS and a camera on Black Merino sheep in Mitra-PT to understand time- and scale-dependent grazing behavior variables related to explanatory pasture conditions. This study concluded that the use of cameras is suitable for assessing variations in the behavior patterns of sheep grazing complex pastures. However, battery and cost data were not presented in the study.

To test Bluetooth Low Energy (BLE) technology, ref. [48] develops a low-cost tracking device for goats and sheep. Due to the communication technology, this device can exchange data with the BLE receiver up to 16m and its autonomy can reach up to ten days. However, this communication technology does not guarantee an exchange of messages in large grazing areas.

In order to verify the LoRa’s capacity, ref. [49] developed a prototype that uses GPS and tested it on the same site where this work is being carried out, but did not build a collar to install on small ruminants. Even so, the study showed that these technologies are sufficient for acquiring and transmitting grazing data. The studies described in refs. [46,50] also used LoRa as communication technology and achieved a communication distance of up to 12 km. This communication technology is, therefore, sufficient for grazing small ruminants in northern Portugal.

Expanding the literature review, recent studies and emerging trends are considered. A thesis on Localization with LPWAN [51] explores the use of LPWAN for localization without GNSS, providing insights into achievable accuracy through real measurements in urban and rural settings. This research highlights the potential of alternative localization solutions using signal characteristics such as Received Signal Strength (RSS), timing measurements, or phase measurements, which are essential for energy-efficient IoT applications. Another study, Low Power Asset Tracking by Means of Narrow Band IoT (NB-IoT) [52], presents a prototype for asset tracking using GPS and NB-IoT technologies. This study compares the precision and energy consumption of both methods, demonstrating the effectiveness of NB-IoT in providing a more energy-efficient solution while maintaining reasonable accuracy for asset tracking. Additionally, research on Narrowband-IoT Network for Asset Tracking System [53] evaluates the implementation of NB-IoT in asset tracking applications. The findings indicate that NB-IoT offers better power efficiency compared to GPRS, with lower power consumption and comparable uplink throughput, making it a suitable choice for IoT applications requiring long battery life and efficient data transmission.

Further expanding on these insights, a comprehensive survey on IoT Positioning leveraging LPWAN, GNSS, and LEO-PNT [54] reviews large-scale and energy-efficient positioning techniques for IoT applications. This survey highlights the limitations of traditional GNSS approaches, such as weak signal propagation in indoor and dense environments and high energy consumption, while emphasizing the potential of alternative positioning solutions. The survey concludes that interoperability between technologies is crucial for enabling energy-efficient communication and positioning in the emerging satellite IoT market. Moreover, a study on designing a low-cost location tracker for IoT applications [55] discusses the development of a cost-effective tracker using GPS/BeiDou and 2G technology, aiming to reduce costs and improve the accessibility for large-scale IoT deployments. This study showcases the successful implementation of the tracker in a pilot test, reinforcing the feasibility of low-cost solutions for IoT asset tracking.

The field of small ruminant tracking encompasses various solutions, each with distinct strengths and limitations. While commercial devices offer practical applications, they often come at high costs and have battery limitations. Academic research is pioneering advanced technologies but faces practical challenges in implementation. Recent studies on LPWAN, NB-IoT, and hybrid positioning techniques highlight the potential for energy-efficient and scalable solutions. Our study aims to bridge these gaps by proposing a low-cost, ergonomic, and long-lasting IoT-enabled solution using technologies such as LoRa and GNSS, addressing the critical needs of small ruminant tracking in silvopastoral systems.

## 3. Objectives and General Requirements

Given the context of the problem, its importance and commercial or non-commercial devices, the main objective of this work is to develop a data acquisition system using IoT technologies to monitor the mobility of goats and sheep during grazing, based on **Autonomy**, **Reliability** and **Ergonomy**.

Bearing in mind the main objective of this work, it is possible to define some specific details of where we want to go with the construction of a system with the aforementioned foundations, so the general requirements of this work are as follows:At least 20 days of operation are expected with a data transfer rate of every 5 min [56].The signal coverage for the device’s real-time communication should reach the radius of a traditional pasture in northern Portugal, i.e., around 2 km.The data to be acquired will be latitude, longitude, time, temperature and humidity of the environment, the relative position of the animal’s neck and the battery percentage of the device.The system must be designed for the animal’s comfort, avoiding rotation around the neck and minimizing any negative impact on welfare.The device must represent the herd with a maximum acquisition error accuracy of 10 m.The device must have the ability to store data in the event of a transmission failure and ensure the effective recovery of stored data.The system must guarantee real-time data flow if all connections are successful.The system must be low-cost, i.e., all parts must be built with low-cost components.

## 4. Device and System Architecture

To ensure the full realization of the stipulated objectives and general requirements, it is crucial that the development of the IoT device and architecture is conducted in a precise and strategic manner. The integration of IoT technologies must be carefully planned and implemented to ensure not only operational effectiveness but also continued relevance in the proposed context.

### 4.1. Sensors

The core technological component of this study was the development of a specialized IoT system designed to capture and transmit detailed grazing data for sheep and goats. This system integrates several high-precision sensors:GNSS L80-M39 (Quectel Wireless Solutions Co., Ltd., Minhang District, Shanghai, China): This sensor is crucial for accurately recording spatial movements. It operates within a temperature range of −40 °C to 85 °C and a voltage range of 3.0 V to 4.3 V. It features 66 acquisition channels and 22 tracking channels. The module consumes 25 mA in acquisition mode, 20 mA in tracking mode, and 1 mA in standby mode. It communicates using the National Marine Electronics Association (NMEA) protocol and takes approximately 35 s to collect the first coordinates in Cold Start mode and less than 1 s in Hot Start mode. The GNSS module provides accuracy with a Circular Error Probable (CEP) of less than 2.5 m, meaning it can determine the location with an error of less than 2.5 m in 50% of measurements when operating autonomously.SHT21 (Sensirion AG, Stäfa, Switzerland): This sensor provides environmental sensing for temperature and humidity measurements. It operates within a temperature range of −40 °C to 125 °C and a humidity range of 0% RH to 100% RH. It features an I2C connection and has a resolution of 0.1% RH humidity and 0.01 °C for temperature. The accuracy is ±2% RH for humidity and ±0.3 °C for temperature. The sensor operates at a voltage range of 2.1 V to 3.6 V and consumes an average of 300 µA, highlighting its low power consumption.ADXL345 (Analog Devices, Inc, Wilmington, MA, USA): This inertial sensor is an advanced inertial sensor designed to detect changes in the animal’s neck by measuring both static gravity and dynamic acceleration. It boasts an accuracy of ±0.5% of the full scale and supports measurement ranges of ±2 g, ±4 g, ±8 g, and ±16 g (±2 g per default). The sensor operates efficiently within a temperature range of −40 °C to 85 °C and requires a supply voltage of 2.0 V to 3.6 V. It features both SPI (3- and 4-wire) and I2C digital interfaces for versatile communication. With a resolution of 3.9 mg/LSB, the ADXL345 can detect inclinations as small as less than 1.0°. In measurement mode, it consumes 23 µA, making it suitable for low-power applications. This combination of features ensures precise and reliable monitoring of the animal’s neck position and movement.

Table 1 summarizes the main characteristics of the device’s sensors, including accuracy, measurement range, resolution and operating temperature.

### 4.2. Control and Communication

The ESP32 microcontroller was used to control the entire device. This microcontroller also has several peripherals and interfaces, including GPIOs, SPI, I2C, DAC and ADC. The model used in the device is the ESP32-WROOM-32E (Espressif Systems Co., Ltd., Shanghai, China), which contains 4 MB of flash memory, 520 kB SRAM, operates at 3.0 V to 3.6 V, has 38 pins GPIOs, and can operate at −40 °C to 105 °C.

For communication purposes, the device was equipped with the LoRa LYLR998 module (REYAX TECHNOLOGY CO., LTD., Taipei, Taiwan), which is renowned for its long-range transmission capabilities and efficient power usage. This technology proves particularly advantageous in vast and diverse grazing areas, where terrain conditions can vary significantly. The LoRa LYLR998 module enables reliable communication over extended distances while conserving battery power, essential for ensuring continuous tracking and monitoring in challenging environmental conditions. Its deployment underscores the robustness required to maintain connectivity and data transmission integrity across expansive agricultural landscapes.

The device was programmed using the Arduino IDE, enabling custom firmware that managed data logging, sensor readings and transmission intervals. This firmware included power management features to optimize battery usage, switching to a low-power mode when data transmission was not taking place, the use of SPI Flash File System (SPIFFS) to store data that was not delivered to the gateway, and the pre-processing and evaluation of acquired data.

The data acquisition interval was set programmatically to ten seconds, aiming to optimize the device performance and generation of a lot of data. In the event of a transmission failure, for example, due to a weak signal or the presence of obstacles, the device has been designed to make one more attempt to send, if it fails to receive confirmation of receipt, it stores data locally in the SPIFFS memory and returns to deep sleep mode, ending a cycle. All messages stored locally are sent in any cycle in which the device manages to communicate with the gateway. The device’s operational block diagram with all the operational states can be found in the link in the Appendix A.

Figure 1 gives an overview of the parts of the device, including the sensors, control, communication and the entire power system. In this figure, you can see the communications technologies used between the microcontroller and sensors and the LoRa module, i.e., which components are at the top of the device and which are at the bottom.

#### Power System

The system’s energy solution combined three Lithium-ion batteries INR18650-35 (Samsung SDI Co., Ltd., Suwon, South Korea) with a high-efficiency solar panel, ensuring continuous operation by harnessing solar energy for battery recharging. Each battery provides 3350 mAh and three units were used, so when charged the system has a charge of approximately 10,050 mAh. These batteries have 4.2 V when fully charged, 3.6 V nominal voltage and stop supplying current at 2.65 V, this information will be fundamental for the development of the battery percentage reading system. This battery model was chosen because of its robustness and charging capacity, which are key characteristics for this application. In addition, they can supply up to 2000 mA, which is more than enough for the system to function, and this will also be covered in more detail in the following sections. The 5.5 V polycrystalline solar panel measures 84.5 × 55.5 mm and has a current of 120 mA with a peak power of 0.66 W when the light intensity is 38000LUX.

All the devices presented above work with a nominal voltage of 3.3 V, so the TPS63020 voltage regulator (Texas Instruments, Dallas, TX, USA) is responsible for guaranteeing 3.3 V to the system. As this module is a booster, it is able to regulate the voltage between 1.8 V and 5 V to 3.3 V, with an efficiency of approximately 96%, in addition to being low-cost. The TP4056 module (Nanjing Top Power ASIC Co., Ltd., Nanjing, Jiangsu Province, China) is used to charge the batteries and integrate them with the solar panel. This module manages the system’s energy, allowing the panel to recharge the batteries and this pair to supply energy to the voltage regulator. A 125 V on/off switch was added, this switch supports up to 3 A and 125 V, which is more than enough for the device. It also has a resistor to be switched on or off, which is essential to avoid unexpected state changes. The power supply system is divided between the upper and lower parts of the device. The solar panel and voltage regulator module are at the top, while the batteries, battery charger module and on/off switch are at the bottom, as can be seen in Figure 1.

To estimate the device’s operating time based on the energy consumption of each operating state and the battery capacity, we will use the Equation (Equation 5). Each operating state describes a specific activity carried out by the device: *deep sleep*, *wake up*, *acquire data*, *send data* and *wait for confirmation*. The execution of all these primary states makes up a complete device operating cycle.

It is important to note that any other operating states are derived from these main ones. The Equation (Equation 5) will be used to calculate the total operating life of the device based on the sequence and duration of each operating state, also taking into account the battery capacity.

To calculate the number of operating cycles in a given period (in hours), we use the equation:(1)$$C_{y}=\;\frac{T}{\sum_{i=1}^{n}t_{i}}$$
where:$C_{y}$ indicates the number of possible operating cycles.*T* represents the operating time (in hours).*t* refers to the time (in hours) consumed by each operating state of the device.

To calculate energy consumption per cycle, we use a summation formula involving different device states:(2)$$C_{c}=\sum_{i=1}^{n}\left(C_{i}\cdot t_{i}\right)$$

Onde:$C_{c}$ represents the consumption per cycle (in mAh).*B* represents the energy consumption (in mA) in each operating state.*t* refers to the time (in hours) elapsed from each operating state.

To calculate the total life of the device, we add the battery capacity in mAh. To simplify the calculations, we assume a linear discharge of the battery, disregarding other factors such as temperature, discharge rate, life cycle, solar panel influences and other operating states, such as data storage if it cannot connect to the gateway. Therefore, the relationship between battery capacity and device consumption can be described as:(3)$$B_{a}t\;-\;C_{y}\cdot C_{c}\;=\;0$$
where:$B_{a}t$ indicates battery capacity (in mAh)

Expanding this equation so that *T* is a function of the cycles and consumption of each cycle, we have:(4)$$B_{a}t\;-\;\;\frac{T}{\sum_{i=1}^{n}\;t_{i}}\cdot \sum_{i=1}^{n}\;\left(C_{i}\;\cdot \;t_{i}\right)\;=\;0$$
(5)$$T\;=\;\frac{B_{a}t\;\sum_{i=1}^{n}\;t_{i}}{\sum_{i=1}^{n}\;\left(C_{i}\cdot t_{i}\right)}$$

Figure 2 shows the developed device prototype, including electronic components and their arrangement in the collar that is put on the animal. The figure illustrates both the upper part, which contains the sensors and modules, and the bottom part, which houses the battery system. The total weight of the device is 550 g.

### 4.3. IoT Architecture

In this context, an IoT architecture was designed, as illustrated in Figure 3, which contains several layers, namely sensing, network, service and interface layers, ensuring the flow of acquired data from its acquisition on the device to its presentation to the end user.

The sensing layer is responsible for collecting data from the animal. A developed data collection device is responsible for collecting the data and sending it to the network layer, which uses LoRa technology since the grazing areas are large and the device has to work for long periods of time (exploring its capabilities to send data over long distances with low power consumption). The network layer responsible for forwarding the data to the cloud application using a gateway containing LoRa and Wi-Fi was developed. The choice of Wi-Fi as the technology for connecting to the Internet provides the practicality and versatility of a wireless connection, making it easy to install and relocate.

The communication protocol chosen to forward the data to the cloud was Message Queuing Telemetry Transport (MQTT). This protocol is widely adopted in Internet of Things (IoT) applications, especially for agricultural applications, due to its optimized characteristics for devices with limited resources and networks with low bandwidth [57].

MQTT is of the publish-subscribe type, which in this case facilitates asynchronous communication between gateway and brokers. This communication model allows the gateway to publish messages for specific topics, and any devices subscribed to those topics receive the messages in real-time. This eliminates the need for direct point-to-point connections and reduces communication overhead, ensuring that a large number of gateways can work in parallel. In addition, MQTT is designed to be lightweight, which means it consumes less of the gateway’s power and processing resources, a critical consideration in this type of application, enabling the system to be scalable.

The service layer, running by an application developed in Node-RED (version 4.0.2), is responsible for receiving, filtering, analyzing, generating alerts, and forwarding the data to the database. Node-RED applications are often used in IoT projects due to their intuitive, flow-based graphical interface, which simplifies programming and connecting different components and services. In addition, Node-RED has a vast library of nodes that support a variety of protocols and services, facilitating integration with APIs, physical devices, databases, and cloud services. Also in the service layer, the InfluxDB database (version 2.7.8) was chosen because it is optimized for storing and querying time series data, which will be used in this work. Secondly, it has high performance in terms of writing and reading large volumes of data, ensuring that even systems with a high frequency of data can function efficiently.

The interface layer is responsible for displaying all the data, alerts and other information about the device. For this purpose, applications were developed in Grafana (version 11.1.3) and Telegram (version 5.4.1). Grafana offers advanced data visualization possibilities, integration with various databases and high performance and scalability, which makes it efficient for this work. On the other hand, the bot developed with Telegram is responsible for receiving alerts and enabling the user to check the latest data collected by the device quickly with their cell phone.

## 5. Deployment of the System

With the architecture defined and structured, it was possible to develop the components of the layers and plan the system’s testing phase more precisely and effectively. In order to carry out these tests, a specific case study was defined, which made it possible not only to monitor but also to evaluate in detail the functioning of each layer of the architecture. This case study served as a representative model of actual operating conditions, providing a clear and comprehensive view of how each system component interacts and performs its functions.

In addition, the use of this case study has made it easier to identify possible points of failure or areas in need of improvement, ensuring that the system as a whole works in an integrated and efficient manner before it is implemented in a production environment. In this way, the testing phase becomes a crucial stage in validating the robustness and reliability of the proposed system.

### 5.1. Data Transmission and Processing

A gateway with the same LoRa module and 2.8 dBi antenna was developed to receive the data transmitted by the device to maximize reception range. This gateway was connected to a local network and configured to forward the received data packets to a service layer via the MQTT protocol, ensuring a secure and reliable data flow.

The LoRa module has been set to a frequency of 868 MHz. Spreading Factor (SF) 9 provides a good relationship between communication range and data rate. Code Rate (CR) 4/5 improves the robustness of communication with the minimum redundancy required for error correction, this CR setting in LoRa means that for every 4 bits of useful data, 1 additional bit is transmitted for error correction, resulting in good error correction capacity and a balance between communication robustness and data rate. The 125 kHz bandwidth optimizes receiver sensitivity and communication range, minimizing interference.

As well as forwarding the data to the MQTT broker, it is also responsible for sending a confirmation message back to the device. This is important for the device to understand that its message has been forwarded to the internet, i.e., there is no need to send it again or save it for future attempts.

The architecture for receiving data published via MQTT was built locally, using dedicated servers for hosting and management. This local setup provided scalability and reliability to handle the data volume and processing requirements. The InfluxDB database was installed on the same local infrastructure, optimized for time series data, which facilitated efficient storage and query performance for the large datasets generated by the device.

### 5.2. Data Visualization and Analysis

Customized dashboards were developed with the Grafana application to display real-time and historical data on grazing patterns, environmental conditions, and device status. These dashboards allowed data to be explored at various formats, from animal movements to aggregated patterns across the entire herd. Each piece of data is presented in one or more panels, Figure 4 shows all the panels built. Within the visualization application developed in Grafana, a heat map panel was developed, displaying the areas preferred by the animals in a colorful and intuitive way, ranging from blue to red, i.e., from the least to the most repeated, this was positioned at Figure 4a.

To visualize the temperature, humidity and speed data, numerical panels were developed with the latest recorded data, which can be seen in Figure 4b–d. The average, according to the period chosen by the user, of each of these data is also shown in Figure 4e–g. The panel located in Figure 4h shows the percentage of the device’s battery, which ranges from 100% (fully charged) to 0% (fully discharged). Finally, Figure 4j shows a panel with a serial graph of the values collected for temperature, humidity, speed and battery percentage. This allows for better analysis and comparison of the history of these data.

To display the relative position of the animal, five positions were mapped to the device, i.e., the X, Y, and Z values are transformed into the values 1, 2, 4, 6, and 8, which represent the straight, leaning forward, left, back and right positions, respectively. This position is relative to the animal, i.e., when the animal leans over to eat, it equals the number 2, for example. When the X, Y, and Z values are not recognized, the value 0 is recorded. These values are displayed serially on a panel positioned at Figure 4k. In addition, the application displays the Wi-Fi and LoRa signal quality of the device/gateway combination in dBi on serial graphics, these graphs can be seen in the panels available in Figure 4l,m, respectively. This information is very important for the user to assess the quality and performance of the device and gateway. Additionally, on the right-hand side of the application, in Figure 4n there is a green LED to indicate the status of the gateway, which turns green when it is working correctly and red otherwise.

There are also two text fields that display the alerts and the type of data being collected. Alerts are displayed with their trigger times, alerts about the animal’s position, temperature, battery percentage and lack of communication between the device and the gateway are generated, these alerts should be displayed in the panel located in Figure 4i. The panel located in Figure 4o shows the type of data collected, indicating whether it was collected in real-time or retrieved from the device’s memory. When the data are retrieved from the device’s memory, the time of collection is displayed.

Grafana searches the database according to the settings defined by the user. On the dashboard, there is a field for defining the search period, for example between days n and n-30, “last 7 days” or “last 30 m”. In addition, the user can set the update frequency, which can be “auto” (where Grafana adapts the update to the set period) or at fixed intervals of 5 s, 10 s, or 15 s. These two settings ensure that Grafana updates your dashboards with a minimum frequency of 5 s, which is more than enough for this type of application.

In addition, a bot has been created on Telegram for quick access to the latest data collected by the device. All alerts are sent to Telegram, and when requested by the user, it displays the latest data on location, temperature, humidity, speed, and battery percentage. This ensures that the user has one more way of accessing the data. The location is returned in the form of a map, meaning that the user can use their cell phone to travel to the herd if necessary.

Integration with Telegram is guaranteed by the application in Node-RED, which features specific Application Programming Interfaces (APIs) for this type of communication. When the user requests data from Telegram, it goes to Node-RED, which searches the database and returns it to Telegram. However, when alerts are generated by Node-RED, they are sent in parallel to the database (so that they can be displayed in the Grafana application) and forwarded to Telegram, i.e., no user action is required.

### 5.3. Device on the Animal

A 3D-printed case made from PLA, designed for durability, weather resistance, and minimal animal discomfort, housed the electronics of each tracker. The case design also included a quick-release mechanism for easy attachment and removal from the collars, which were adjustable to accommodate various animal sizes. Figure 5 shows the collar that supports the device already installed on the animal.

### 5.4. System Installation Location

The system was implemented and tested at the Polytechnic Institute of Bragança (IPB) (41.7947° N, 6.7685° W) has a small ruminant herd of around fifty animals, mostly goats, and grazes them daily. These animals graze in some defined areas within the campus, Figure 6 shows a map containing the stable where the animals are concentrated and the areas belonging to the IPB where the herd grazes. During the grazing days, two test areas were divided just to organize the tests, which can be seen in pink and yellow, named zones one and two, respectively. In addition, the gateway was positioned in the animals’ stable, where the animals concentrate before and after all the grazing. Zone one is approximately 73.350 m^2^ and the furthest point from the stables is approximately 720 m. Zone two is approximately 33.650 m^2^ and its furthest point is approximately 570 m. Both zones have open countryside, but between zone one and the stable, there are large buildings that can interfere with communication between the device and the gateway, while zone two does not have so many obstacles. These details can be seen in Figure 6.

The device must be attached to an adult animal already adapted to tracking collars, which is defined by the people responsible for the herd. The gateway, on the other hand, must be attached in a place close to the stable.

### 5.5. Software and Code Availability

The custom firmware (version 1.0) developed for the GNSS trackers and configuration scripts for the gateway and local processing pipeline has been made available under an open-source license on GitHub to ensure transparency and facilitate replication, available in the Appendix A. This repository includes comprehensive documentation on system setup, sensor calibration procedures, and data analysis examples, providing a complete guide for researchers and practitioners interested in deploying similar IoT-based monitoring systems in their work.

### 5.6. Ethical Considerations

Detailed attention was paid to ensure that the technological interventions did not cause harm to the animals. We vetted the tracker design for comfort and safety and obtained ethical approval from the relevant institutional review board. Regular checks ensured the technology did not adversely affect the animals’ natural behavior or well-being.

## 6. Results

Implementing an IoT-based solution for monitoring sheep and goat grazing patterns has produced a comprehensive set of data, shedding light on the operational efficiency of the deployed equipment and its practical applications in ecological monitoring. The experiment was meticulously designed to assess the system across multiple dimensions: data transmission reliability, battery longevity and operational efficiency, sensor accuracy, and overall system robustness under various field and animal conditions.

The tests were carried out in a practical way, i.e., the collar with the device was put on random adult goats that were already used to collars and they grazed normally. On each day of grazing, only one goat received the device. This was repeated twice for each zone on different grazing days, i.e., there were four grazing days and it was possible to monitor the entire functioning of the system.

### 6.1. Data Transmission Reliability

The system’s robust data transmission capacity was the cornerstone of the study’s success. Table 2 shows the data collected during the four tests, where it is possible to analyze the correlation between each test and the amount of data collected and whether it was captured in real-time or had to be stored due to a transmission failure. It can be seen that the average time in zone two for the device to collect and send data is around 30 s, i.e., discounting the 10 s in sleep mode, it takes 20 s for the device to collect and send data. This time increases to an average of 72 s for zone one and 62 s if we disregard the time consumed by deep sleep mode. The time taken to collect and send data is influenced by the time taken to collect data from the GNSS and the number of attempts to send data.

The tests carried out in zone two showed an average of 2.5× more data/hours than in zone one due to the obstacles in the zone, which prevent real-time communication and force the device to make more attempts to send and/or save data to send in the future. In particular, LoRa technology facilitated consistent long-range data transmission, proving its suitability for remote pasture monitoring applications. Its coverage was sufficient to cover both grazing zones, even with the buildings between zone one and the gateway. Even so, it was possible to establish communication between the device and the gateway at the furthest point, i.e., 720 m. A test was carried out to check the range of the LoRa module without interference from physical obstacles and 6 km was achieved with −103 dBi, meaning that it would still be possible to reach greater distances with this module.

This result is sufficient for this application case, but for grazing sites where there is no physical demarcation and the vegetation and terrain are varied, more tests need to be carried out to better understand the efficiency of using LoRa as a communication technology. In addition, the GNSS Hot Stard mode accelerated the data acquisition process.

### 6.2. Device Lifetime and Efficiency

Another highlight was the superior battery performance, which significantly exceeded initial projections. The system’s power supply has shown promise in terms of battery life due to its storage capacity and the contribution of the solar panel. The bench tests showed that, with *deep sleep* in 300 s, the device could reach 37 days of uninterrupted operation. Figure 7 shows the correlation between days of battery life and data acquired according to the deep sleep mode selected, ranging from 30 s to 300 s.

Calculate the lifetime, an average of the consumption (in mAh) of the different operating states of the device was collected on the bench and then the time that the device can remain active was estimated according to the capacity of the battery pack.

An average of the consumption of the *deep sleep*, *wake up*, *acquire data*, *send data* and *wait for confirmation* states were collected on the bench. Table 3 shows the values acquired on the bench. All states have an average duration and consumption; however, in order to evaluate the consumption of the device as a function of the time in *deep sleep*, it was varied between 30 s and 300 s.

We then applied the Equation (Equation 5) which determines the lifetime of the device as a function of the battery capacity and *deep sleep*. So we have
(6)$$T\;=\;\frac{B_{a}t\cdot \left(t_{Wa}+t_{A}+t_{S}+t_{Wc}+t_{Dl}\right)}{Wa\cdot t_{Wa}+A\cdot t_{A}+S\cdot t_{S}+Wc\cdot t_{Wc}+D_{l}\cdot \;t_{D_{l}}}$$

Onde:$W_{a}$ Wake up.*A* Acquire data.*S* Send data.$W_{c}$ Wait for confirmation.$D_{l}$ Deep sleep.

Atribuindo os valores de cada variável temos a seguinte relação:(7)$$T\;=\;\frac{10,050\cdot \left(3+18+0,4+4+t_{Dl}\right)}{65\cdot 3+45\cdot 18+150\cdot 0,5+46\cdot 4+8\cdot t_{D_{l}}}$$
(8)$$T\;=\;\frac{10,050\cdot \left(25.4+t_{Dl}\right)}{1249+8\cdot t_{D_{l}}}$$

Therefore, with the Equation (Equation 8) it is possible to define the time (in hours) of the device’s lifetime. For better visualization, Figure 7 shows the lifetime and the number of possible cycles in the defined time of the *deep sleep* state. The Equation (Equation 1) was used to calculate the number of cycles.

The system’s energy-efficient design by incorporating a solar panel extends the battery life, ensures continuous operation, and minimizes the monitoring system’s environmental footprint by leveraging renewable energy sources. During the grazing tests, the power supply system showed the data shown in Table 4.

All four tests were carried out on sunny days, which contributed to the panel supplying power to the batteries. In the grazing tests, the solar panel had an influence on the discharge of the battery, by calculating the average discharge in the tests we arrived at 0.69% per hour.

This consumption can be reduced by adapting the deep sleep time to the grazing patterns of each herd. As each herd has a different pattern, i.e., the start and end times of grazing vary according to the time of year and the shepherd’s decisions, it is not possible to set fixed times when the device increases or decreases the deep sleep mode. This can be conducted in future efforts, with individual herd notions. This way, while the animals are in the stable, the device will not collect unnecessary data.

### 6.3. System and Sensor Accuracy, Robustness and Adaptability

The accuracy of the GNSS locators in relation to established geospatial references was rigorously evaluated. The GNSS trackers, essential for understanding grazing dynamics, provided high-resolution data on movement patterns, grazing hotspots and resting places. This level of detail is key to quantifying impacts on the ecosystem and informing sustainable grazing management practices.

The GNSS has two types of drives, Cold Start and Hot Start. Whenever the GNSS is switched on after three hours of being switched off or in standby mode, it is triggered by Cold Start, while if it has been in standby mode for the last three hours, it is triggered by Hot Start. In practical tests, it was found that as soon as it was switched on in Cold Start mode, the first longitude and latitude data collected by the GNSS showed a greater error than expected, causing variations of up to 100 m, while when it was switched on in Hot Start mode, this error decreased considerably, varying the position by up to 15 m. To solve this, the firmware discards the first 15 points collected when the GNSS is switched on in Cold Start and the first 3 points collected when switched on in Hot Start. With this process, the error now varies up to 10 m for both types of trigger. As the points collected represent the herd, the error accepted by the GNSS is acceptable, even in adverse vegetation conditions, where the error tends to be intensified.

In addition to the performance of each individual component, the study tested the overall robustness of the system and revealed no significant loss of performance or data quality, highlighting its adaptability and resilience. This robustness ensures that the system can be used in various terrains and animal conditions, making it a versatile tool for ecological studies worldwide.

### 6.4. Experimental Interpretation

The experiment’s findings confirm the viability of IoT-based systems for pastoralism and rangeland monitoring. The high data transmission reliability ensures the collection of comprehensive datasets with minimal data loss, a critical factor for longitudinal studies. The exceptional battery life extends the operational window of the devices, reducing maintenance intervals and operational costs. The accuracy of the sensors provides the granularity needed to analyze complex ecological interactions and behaviors. Finally, the system’s robustness guarantees reliable performance across various field and animal conditions.

The rest of the data were presented as expected by the graphical user interface created in the Grafana software. Figure 8 shows an overview of the dashboards with data from the second test carried out in zone two. Here, the user can see the heat map generated with the animals’ preferred locations, the temperature, humidity, battery percentage, and speed in numerical format (the last value collected and an average) and the serial graph.

There is also the relative position of the animal’s neck and the LoRa signal quality data generated between the device gateway and Wi-Fi between gateway-Internet in dBi. As no alerts were generated in this test, the alert panel is empty. There is also a record of the last data points types collected, which in this case, were in real-time.

Collectively, these results validate the IoT system’s technical capabilities and underscore its potential to revolutionize ecological monitoring and herding management. By facilitating detailed and reliable data collection on livestock grazing patterns, the system provides a foundation for valuing the ecosystem services rendered by silvopastoral systems, paving the way for more sustainable practices and conservation strategies.

## 7. Discussion

The deployment of an IoT-based solution for the detailed monitoring of sheep and goat grazing patterns represents a pivotal step forward in integrating technological advancements with rangelands and landscape research [48,58]. This integration is crucial for developing sustainable silvopastoral practices that support rather than undermine ecosystem services [49]. The robust performance of IoT devices, evidenced by their resilience and reliability across various terrains and animal conditions, validates the effectiveness of such technology in ecological monitoring tasks. Unlike commercial solutions that focus on real-time flock tracking, this study uniquely emphasizes documenting the shepherd’s role, providing a low-cost solution to acknowledge and fairly remunerate their contributions. This approach not only enhances productivity and income but also promotes the conservation of ecosystem services [59,60].

The implications of our findings extend beyond the validation of IoT technology for silvopastoral monitoring. This research contributes significantly to the broader discourse on sustainable rangeland management and its role in preserving ecosystem integrity by providing a granular view of grazing patterns. The nuanced understanding facilitated by our study echoes recent scholarly work that underscores the importance of data-driven approaches to silvopastoral sustainability [61,62]. It becomes clear that leveraging IoT technologies can bridge the gap between traditional silvopastoral practices and the need for nature conservation, offering a pathway to reconciling traditional livestock productivity with ecological and landscape stewardship.

Technological advancements such as improved battery life, stable electrical connections, and miniaturization of electrical components have facilitated the use of sensors to remotely monitor animal behavior [63]. Previously, livestock location data were usually stored on the device, but the development of IoT technologies has allowed the development of real-time and near real-time sensor devices [17,64,65].

The total cost of the set is around 120–180 euros, similar to other private initiatives or similar work [40,64]. This price was achieved by selecting low-cost electronic components. On the other hand, the cost of the collar that supports the device was high because it is a work of art, which could be solved with a larger scale of production.

Despite these promising results, it is essential to recognize the limitations inherent in this study. The capabilities of the devices used restricted the range that the IoT configuration could capture, which is a notable limitation. This limitation is due to the conditions of the grazing site and the limits of LoRa technology, which can have higher signal losses than technologies such as NB-IoT and Sigfox [66].

To alleviate the problems arising from communication between the device and the gateway, a delivery confirmation system was used, similar to the one used in the LoRaWAN protocol, where the device sends data to the gateway and waits for it to return a confirmation that everything went well. As with LoRaWAN, this mechanism can cause problems with data transmission, but it is a good practice for LPWAN [67,68,69]. Additionally, a memory system has been added, where data not delivered to the gateway are stored and present at a later opportunity. This system ensured that 98% of the data collected passed through all the layers of the architecture, but further research is needed into the security of the stored data and into the capacity, performance, and lifetime of the microcontroller when using this type of peripheral.

This limitation points to a gap in current research and suggests the potential for incorporating a wider range of self-powered, internet-connected gateways to cover more remote landscape conditions due to terrain relief, vegetation structure, and built structures on the ground. This would increase the availability of the dataset, allowing faster recognition and response by the herder to avoid prohibited grazing locations and times [70].

To optimize data acquisition from the device and the orientation of the solar panel, the counterweight generated by the lower part of the collar means that the box supporting the sensors remains on top of the animal, i.e., at its highest point, which provides an efficient inclination of the GNSS and LoRA modules [64]. In ref. [71] batteries were also used as a counterweight to hold an accelerometer above the necks of cows. This also maximizes solar incidence on the solar panel, ensuring that it makes a greater contribution to charging the batteries.

The use of GNSS for tracking sheep and goats is efficient in various scenarios [47,72,73,74]. Even so, there is a propagation of errors that can arise depending on the application, especially in application conditions that present physical barriers [64]. However, these flaws can be improved through quality control methods [75] and/or integration with other geolocation acquisition methods, based on LPWAN, for example [51,54].

Moreover, the geographical scope of this study was limited, focusing on specific Iberian Mediterranean mountain rangelands that may not fully represent the global diversity of grazing practices and ecological contexts. Expanding the deployment of the IoT framework to encompass various ecological zones, livestock species, and silvopastoral systems would enhance the generalization of the findings, providing insights applicable to a broader array of landscape and ecological settings.

For the continuation of this work, improvements should be made to the device, such as the inclusion of water resistance and optimization of the collar, in order to make its fit to the animal more optimal. Reducing one of the batteries and adding a solar panel should also be considered, as this would make the set lighter and reduce its price. In addition, the confirmation protocol should be improved to identify replicated data.

Future research should aim to scale the IoT infrastructure to include a broader spectrum of herding conditions and extend its application to a more comprehensive range of silvopastoral systems and ecological contexts [76,77]. There is also a critical need to develop sophisticated analytical models capable of synthesizing the high-resolution data collected via IoT devices. Employing advanced data analytics and machine learning techniques could uncover patterns and correlations that are not immediately apparent, offering more profound insights into the ecological impacts of grazing. Such analytical advancements could inform predictive models that assess the implications of various grazing strategies on ecosystem services, guiding the development of sustainable herding management practices.

## 8. Conclusions

The system presented in this document, based on LoRa communication and other IoT technologies, has proven effective in monitoring small ruminant herds. Even though it was a prototype, the cost of the device remained lower than that of other commercial devices. The device’s data storage system was efficient in situations where connection to the rest of the architecture was not possible. Additionally, the integration between the solar panel and the three batteries allowed uninterrupted operation for up to 37 days with 5-min interval acquisitions, showing a promising device lifetime. All objectives and system requirements have been met.

Experimental results demonstrated the high performance and robustness of the IoT system, with minimal data loss and significant battery efficiency, validating its suitability for long-term field evaluations. LoRa technology ensured consistent communication over long distances, covering the entire grazing zone and a range of 6 km in open areas. The GNSS module provided high-resolution data on movement patterns, with an accuracy of up to 10 m after firmware adjustments. The two-part division of the device ensured it did not rotate on the animals’ necks, demonstrating adaptability and resilience in various terrains and animal conditions.

This work uniquely focuses on documenting the shepherd’s role in the ecosystem, distinguishing itself from commercial solutions that emphasize real-time flock tracking. By offering a low-cost prototype that imposes no financial burden on the shepherd, it aims to recognize and remunerate their contributions, thereby enhancing productivity and ecosystem services. Further research is needed to enhance various aspects of the device and the system as a whole. This effort requires a multidisciplinary approach, incorporating insights from technology, pasture science, and sustainable pastoralism to develop strategies that promote productivity and ecological well-being.

## Figures and Tables

**Figure 1 sensors-24-05528-f001:**
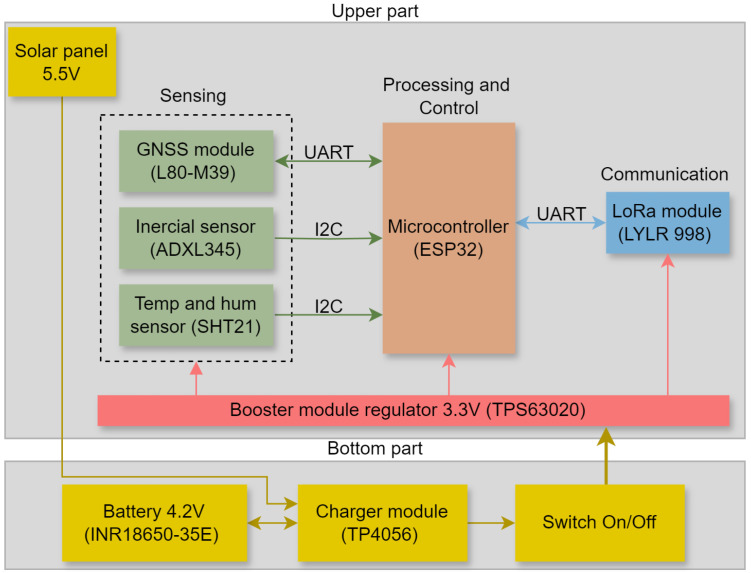
Device overview. The upper part is responsible for supporting the sensors and data acquisition and transmission modules, as well as a microcontroller for processing and control. The bottom part is made up of the system’s power supply.

**Figure 2 sensors-24-05528-f002:**
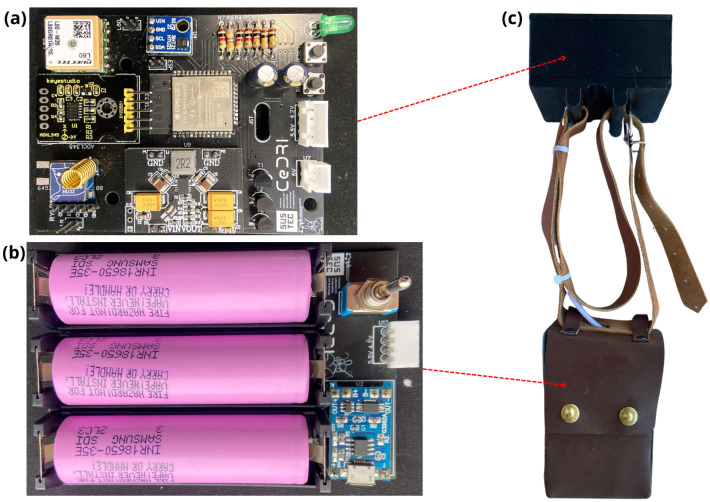
Device plates and collar: (**a**) Top plate; (**b**) Bottom plate; (**c**) Collar with top and bottom parts already attached to their housings.

**Figure 3 sensors-24-05528-f003:**
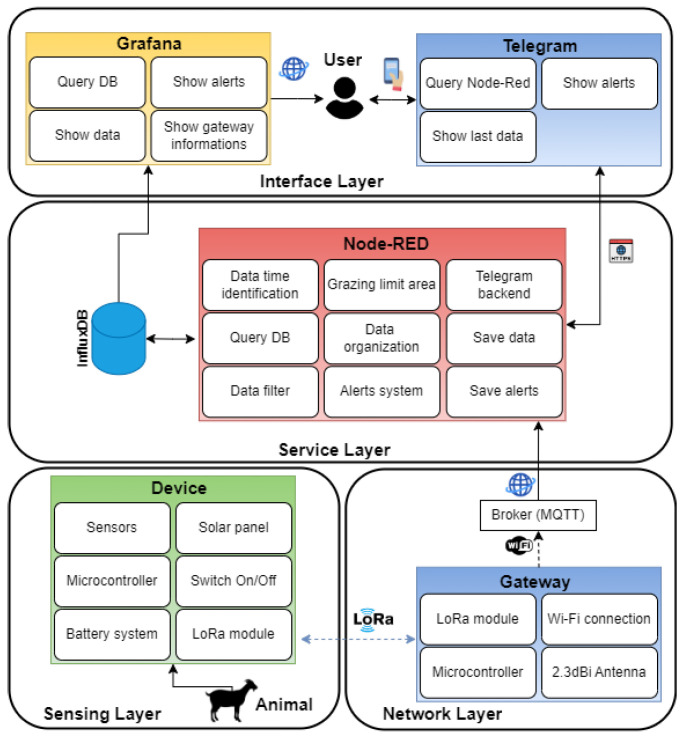
Overview of IoT architecture, including the acquisition, transport, service and interface layers.

**Figure 4 sensors-24-05528-f004:**
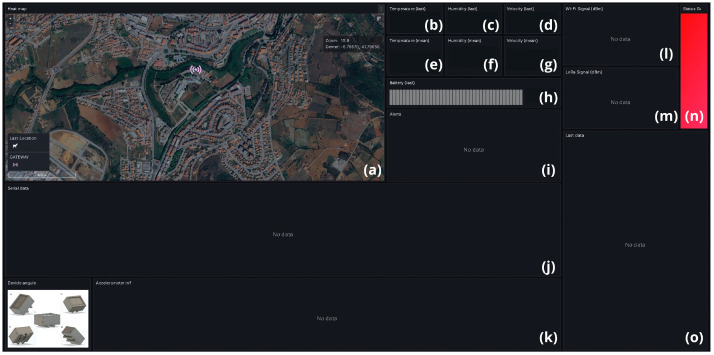
Layout of the panels built at Grafana: (**a**) Heat map; (**b**) Temperature; (**c**) Humidity; (**d**) Speed; (**e**) Average temperature; (**f**) Average humidity; (**g**) Average speed; (**h**) Device battery; (**i**) Alerts field; (**j**) Temperature, humidity, speed and battery percentage in serial graph; (**k**) Animal neck position; (**l**) Wi-Fi signal quality; (**m**) LoRa signal quality; (**n**) Gateway status; (**o**) Type of data received (real-time or memory).

**Figure 5 sensors-24-05528-f005:**
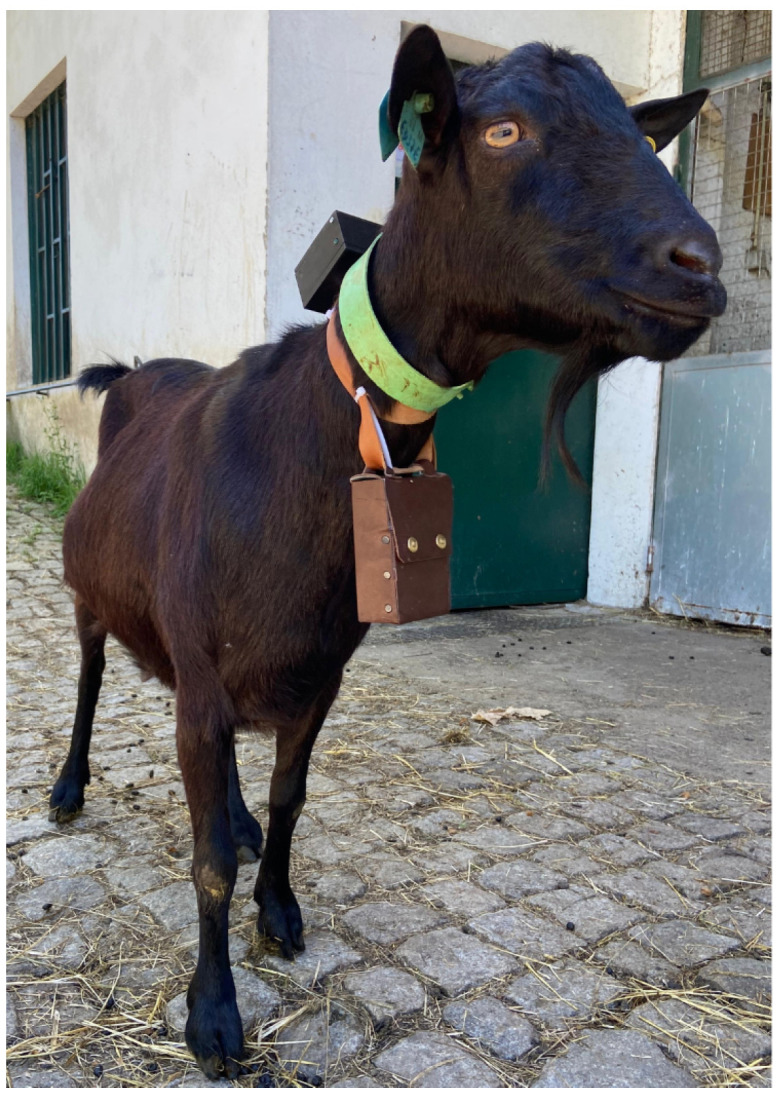
Developed IoT collar on the animal.

**Figure 6 sensors-24-05528-f006:**
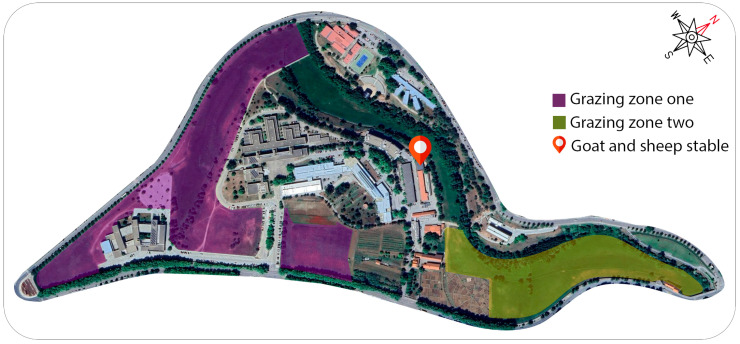
System installation location with the grazing zone and stable.

**Figure 7 sensors-24-05528-f007:**
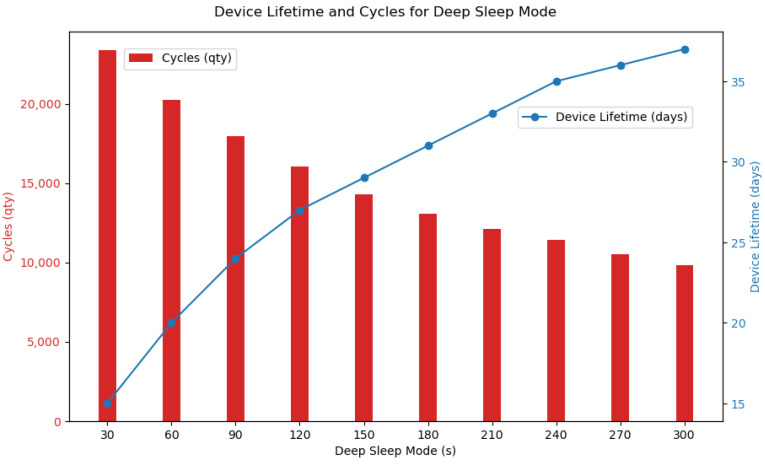
Relationship between device lifetime and device cycles based on programmed *deep sleep*.

**Figure 8 sensors-24-05528-f008:**
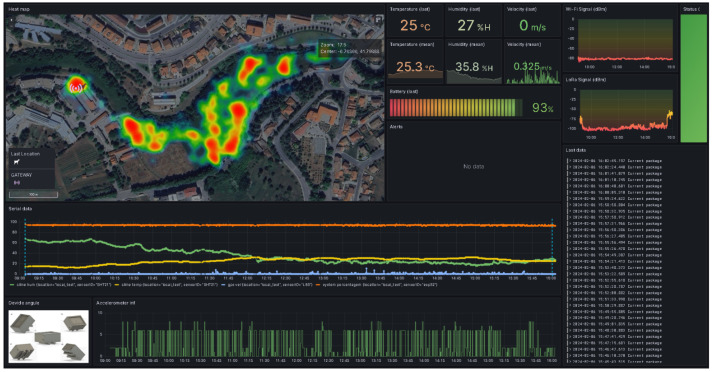
Overview of the acquired data being presented by Grafana. Each panel shows different data, from positional data to system-generated alerts.

**Table 1 sensors-24-05528-t001:** Specifications of the IoT device sensors.

Sensor	Accuracy	Measurement Range	Resolution	Operating Temperature
GNSS L80-M39	CEP < 2.5 m	Global	-	−40 °C to 85 °C
SHT21	H: ±2% RH, T: ±0.3 °C	H: 0% RH to 100% RH, T: −40 °C to 125 °C	H: 0.7% RH, T: 0.01 °C	−40 °C to 125 °C
ADXL345	±0.5%	±2 g, ±4 g, ±8 g, ±16 g	3.9 mg/LSB	−40 °C to 85 °C

**Table 2 sensors-24-05528-t002:** This table shows a list of the tests carried out and the number, grazing time, number, type (real or memory time) and, most importantly, the amount of data collected per hour.

Test	Zone	Start (h)	Finish (h)	Collected Data (qty)	Real Time Data (qty)	Memory Data (qty)	Data/Hour (qty)
1	1	13:17	16:28	129	15	114	40
2	1	13:00	16:01	193	64	129	64
3	2	13:03	15:16	274	215	59	123
4	2	09:04	16:03	876	861	15	125

**Table 3 sensors-24-05528-t003:** Consumption of each device process. The time of the deep sleep process has been omitted because its variation will be explored.

State	Consume (mA)	Time (s)
Wake up	65	3
Acquire data	45	18
Send data	150	0.4
Wait for confirmation	46	4
Deep sleep	8	~

**Table 4 sensors-24-05528-t004:** Results of variation in battery percentage throughout the grazing tests.

Test	Zone	Time (h)	Start (%)	Finish (%)	Variation/Hour (%)	Total Variation (%)
1	1	02:45	93.85	94.20	+0.12	+0.35
2	1	03:01	93.65	93.10	−0.18	−0.55
3	2	02:13	95.00	93.00	−1.81	−2.00
4	2	06:59	94.05	93.45	−0.08	−0.60

## Data Availability

Data are contained within the article.

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
