# Peer review of "Development of an IoT-Based Device for Data Collection on Sheep and Goat Herding in Silvopastoral Systems"

_sensors, 2024, doi:10.3390/s24175528_

Round 1

Reviewer 1 Report

Comments and Suggestions for Authors

The authors conducted a practical study of the design and evaluation of IoT nodes/networks for livestock monitoring, assessing the QoS and energy (estimating the lifetime of the IoT devices) performance. The paper is well-written but has a few limitations, as pointed out below. 

1. The authors have not stated their original contribution, which makes it difficult to know what the contributions of the authors are. I recommend that the authors should introduce a paragraph (before the last describing the outline of the work) to outline the contributions of this manuscript in bullet points or numbering. 

2. Figure 6 shows a relationship between sensor data, deep sleep mode, and battery life.  The relationship between deep sleep mode and battery lifetime seems logical as the deep sleep mode increases (the device sends most of its time in deep sleep, consuming relatively little energy), the lifetime of the device (the time required to consume all the energy stored in the battery of the device) increases. I will rather prefer the term lifetime of the device rather than battery's lifetime as that has a different meaning and is misleading in this context.  See the following paper:

- Kuaban, G. Suila, E. Gelenbe, T. Czachórski, P. Czekalski, and J. Kewir Tangka, "Modelling of the Energy Depletion Process and Battery Depletion Attacks for Battery-Powered Internet of Things (IoT) Devices", sensors, vol. 23, issue 6183, 07/2023.

However, the authors should explain how they obtained the lifetime of the IoT nodes
shown in Fig. 6. It is not clearly explained.

3.  The conclusion should reflect the findings from the experiments. That is, it should be deduced from the studies' outcomes. At the moment, the conclusion seems general and should be improved. 

Comments on the Quality of English Language

The language quality is acceptable. 

Author Response

Dear reviewer, I would like to sincerely thank you for the valuable comments and suggestions you provided when reviewing my manuscript. Your comments were extremely useful and helped to improve the quality of the work.

All the yellow comments in the article are new changes. In response to the points raised, I present below a detailed description of the changes made and the necessary justifications:

Comments 1: The authors have not stated their original contribution, which makes it difficult to know what the contributions of the authors are. I recommend that the authors should introduce a paragraph (before the last describing the outline of the work) to outline the contributions of this manuscript in bullet points or numbering.

Response 1:  There is a section called “Author Contributions” after the conclusions of the work. Here you can see each author's contribution. This can be found between lines 689 and 691. 

Comments 2: Figure 6 shows a relationship between sensor data, deep sleep mode, and battery life.  The relationship between deep sleep mode and battery lifetime seems logical as the deep sleep mode increases (the device sends most of its time in deep sleep, consuming relatively little energy), the lifetime of the device (the time required to consume all the energy stored in the battery of the device) increases. I will rather prefer the term lifetime of the device rather than battery's lifetime as that has a different meaning and is misleading in this context.  See the following paper:

- Kuaban, G. Suila, E. Gelenbe, T. Czachórski, P. Czekalski, and J. Kewir Tangka, "Modelling of the Energy Depletion Process and Battery Depletion Attacks for Battery-Powered Internet of Things (IoT) Devices", sensors, vol. 23, issue 6183, 07/2023.

However, the authors should explain how they obtained the lifetime of the IoT nodes
shown in Fig. 6. It is not clearly explained.

Response 2: The suggested article is really good for evaluating the lifespan of a device. To calculate the lifetime, the methodology adopted is presented between lines 275 and 304, where the calculations are based on the collection cycles and their consumption. In the results, the equations are applied and finally it is possible to plot the relationship between cycles and days of lifestime as a function of the deep sleep time adopted, which can be seen between lines 503 and 523. Adding these calculations was of great value to the reader's understanding. The term “Device Lifetime” has been adopted both in the text and in Figure 7 (formerly Figure 6).

Comments 3: The conclusion should reflect the findings from the experiments. That is, it should be deduced from the studies' outcomes. At the moment, the conclusion seems general and should be improved. 

Response 3:  The conclusion was changed from being general to something more direct to the results. The main results discussed in the “results” and “discussion” session were presented, such as LoRa calcance, GNSS accuracy, device lifetime, device ergonomics.

I hope that the revisions meet your expectations and that the revised manuscript now complies with MDPI standards. I remain at your disposal for any further clarifications or necessary adjustments.
Thank you again for the time and effort you put into revising this work.

Sincerely,
Mateus Costa

Reviewer 2 Report

Comments and Suggestions for Authors

The paper presents a very interesting IoT application, and I congratulate the authors for their efforts. While it is a well-contextualized and insightful paper with a highly relevant case example, I believe some improvements in the information provided to the reader are necessary before it can be accepted for publication. I will list a some recommendations for authors to assess the feasibility of including this information in the text of the manuscript.

I believe the title is inaccurate, as it suggests quantification. I recommend reconsidering the title to better reflect the content of the paper.

Before Section 3, where you present the system architecture, it would be beneficial to include a list of project requirements. This will help justify the choice of architecture and assist the reader in understanding your design decisions. These requirements should encompass link coverage, system autonomy, operating temperature, goat position accuracy, goat head inclination, and the minimum required data/hours, among others. You can refer to the literature for guidance on properly describing these requirements.

Figure 1 is very good, as is your explanation. However, I think the explanation of Figure 2 should be placed earlier to show the architecture of the instrumentation device attached to the goat, which you emphasize later in the text. Additionally, the quality of Figure 2 needs to be improved. I would also like to see more electrical details of the power system. Another question I have is whether your system includes storage memory, as you mention data recorded in Section 5.1, specifically in line 355. This information also appears in the "memory data" section of Table 1.

Your GNSS L80 has an error of 2.5 meters, excluding multipath effects, which in rural environments with dense vegetation can further increase this error, as indicated by previous literature. Conducting a static test of the device and calculating the variance of the position error for your application would be important. This would demonstrate to the reader how accurately you can monitor your herd in both clear skies and dense vegetation areas. These issues could be better discussed in Section 5.3.

As a future recommendation, the authors could enhance their system by integrating ADXL345 accelerometer data with GNSS data using a Kalman filter. This integration could significantly improve the quality and accuracy of positioning in dynamic environments, providing more robust and reliable location data for monitoring applications.

It would be beneficial to justify the choice of MQTT for your system, especially considering factors like the number of subscribers and the potential scale of a large herd of goats. While MQTT is efficient and lightweight, other protocols such as NATS, or RabbitMQ might offer different advantages, particularly in scenarios with very large numbers of devices or subscribers. NATS is known for its simplicity and high performance, making it suitable for distributed systems where scalability and low latency are fundanemtal. RabbitMQ offers robust message queuing with support for various messaging patterns, ideal for complex workflows and reliability. Considering these factors, would be appropriate to justify the protocol choice and ensure optimal performance and scalability for monitoring a large herd of goats.

A block diagram explaining the implementation of your device's firmware is very important to show the reader.

In Section 4.2, it would be important to discuss the coverage achieved with LoRa and provide specific details that are currently missing. Here are some points to consider and clarify for the reader: (1) Frequency and Modulation: Specify that you are using the 433 MHz frequency band for LoRa communication. Also, clarify if you are using the default spreading factor (SF) or if you have adjusted it, as this can impact data rate and coverage. (2) Data Rate: Mention the data rate you are achieving with your LoRa setup. This is crucial for understanding the throughput capabilities of your system. (3) Coverage Testing: Describe any tests conducted to evaluate the coverage of your LoRa system. Mention the environment where these tests were performed (e.g., open field, terrain with hills, forested areas) and discuss the maximum range achieved in these tests. This information helps validate the claimed coverage of up to 15 km from the datasheet in real-world conditions. (4) Comparison and Environment: Compare the theoretical range from the datasheet with your practical results. If the maximum range tested was 720 m (as mentioned in Section 4.5), clarify why this distance was chosen and how it reflects real-world deployment scenarios. (5) Challenges and Considerations: Discuss any challenges encountered in achieving the desired coverage, especially in challenging environments such as areas with hills or dense vegetation as you mentined in the discussion of Table 1. This helps contextualize the effectiveness of your LoRa implementation.

In Section 5.1, where coverage is briefly discussed, expand on these points to provide a more comprehensive understanding of your system's performance and the factors influencing coverage.

In Section 4.3, it would indeed be beneficial to provide a detailed overview of your data visualization system, including numerical panels and possibly a visual representation similar to Figure 7. Here are some suggestions on how to address this and clarify your system's functionality: (1) Data Visualization System: Describe the layout and functionality of your data visualization system. Explain how numerical panels are utilized to display key metrics or real-time data from the herd monitoring devices. (2) Integration with Telegram: Elaborate on how your application interfaces with Telegram for messaging. Describe the purpose and use cases for Telegram integration in monitoring a large herd. For example, explain how alerts or status updates are sent to Telegram and how stakeholders can interact or receive notifications via this platform. (3)   Real-Time Data Display: Clarify how your system handles real-time data collection and display. Discuss the frequency of data updates and the types of information presented (e.g., GPS coordinates, temperature). Is there a option to export txt data?

Section 4.4, what is the mass of the system that the goat has to carry around its neck?

In section 5.1 you have an explanation between lines 355-357 about times that I wasn't able to follow, I recommend clarifying. Why do you need 20s to transmit your data, that's not clear to me? Wouldn't it be better to store larger amounts of data to increase the sleep period?

In Section 5.2, it would be valuable to discuss the issue of nighttime operation and consider implementing a strategy where the equipment enters a prolonged sleep mode during periods of inactivity, particularly when the herd of goats is resting.

Section 5.3 needs to be substantially improved, there is no information there that demonstrates accuracy or any other feature.

Conclusion: Recapitulate key findings and contributions of your research. Summarize the performance metrics evaluated and their implications for practical applications.

Author Response

Dear reviewer, I would like to sincerely thank you for the valuable comments and suggestions you provided when reviewing my manuscript. Your comments were extremely useful and helped to improve the quality of the work.

All the yellow comments in the article are new changes. In response to the points raised, I present below a detailed description of the changes made and the necessary justifications:

Comments 1: I believe the title is inaccurate, as it suggests quantification. I recommend reconsidering the title to better reflect the content of the paper.

Response 1: The title has been changed from “Quantifying Ecosystem Services of Silvopastoral Systems: An IoT Approach to Data Collection on Sheep and Goat Herding” to: “Development of an IoT-enabled Device for Data Collection on Sheep and Goat Herding in Silvopastoral Systems”. The new title is more direct and makes it clear that the article is about the development of a device.

Comments 2: Before Section 3, where you present the system architecture, it would be beneficial to include a list of project requirements. This will help justify the choice of architecture and assist the reader in understanding your design decisions. These requirements should encompass link coverage, system autonomy, operating temperature, goat position accuracy, goat head inclination, and the minimum required data/hours, among others. You can refer to the literature for guidance on properly describing these requirements.

Response 2: The “bjectives and General Requirements” section was added and the requirements in the form of items were added. These requirements present the limitations of range, data collected, local data storage, cost, among others. These can be seen between lines 164 and 186.

Comments 3: Figure 1 is very good, as is your explanation. However, I think the explanation of Figure 2 should be placed earlier to show the architecture of the instrumentation device attached to the goat, which you emphasize later in the text. Additionally, the quality of Figure 2 needs to be improved. I would also like to see more electrical details of the power system. Another question I have is whether your system includes storage memory, as you mention data recorded in Section 5.1, specifically in line 355. This information also appears in the "memory data" section of Table 1.

Response 3: The old Figure 2 (now Figure 1) has been improved and placed before the old Figure 1 (now Figure 2), plus the entire explanation of the device precedes the explanation of the architecture. To make this possible, the former section “3. System Architecture” has been renamed “4. Device and System Architecture”. More details about the power supply system have been added, which can be seen between lines 251 and 273. The use of memory is commented on for the first time on line 236 and its use between lines 239 and 246. 

Comments 4: Your GNSS L80 has an error of 2.5 meters, excluding multipath effects, which in rural environments with dense vegetation can further increase this error, as indicated by previous literature. Conducting a static test of the device and calculating the variance of the position error for your application would be important. This would demonstrate to the reader how accurately you can monitor your herd in both clear skies and dense vegetation areas. These issues could be better discussed in Section 5.3.

Response 4: Between lines 209 and 212, the manufacturer's data on GNSS accuracy was presented. Lines 538 to 559 show how the firmware treats the first GNSS points and how this impacts on the accuracy of the points. In addition, the “Cold” and “Hot” initialization modes were presented and how this influenced the results.

Comments 5: As a future recommendation, the authors could enhance their system by integrating ADXL345 accelerometer data with GNSS data using a Kalman filter. This integration could significantly improve the quality and accuracy of positioning in dynamic environments, providing more robust and reliable location data for monitoring applications.

Response 5: We can certainly use this filter in the future, thanks for the suggestion.

Comments 6: It would be beneficial to justify the choice of MQTT for your system, especially considering factors like the number of subscribers and the potential scale of a large herd of goats. While MQTT is efficient and lightweight, other protocols such as NATS, or RabbitMQ might offer different advantages, particularly in scenarios with very large numbers of devices or subscribers. NATS is known for its simplicity and high performance, making it suitable for distributed systems where scalability and low latency are fundanemtal. RabbitMQ offers robust message queuing with support for various messaging patterns, ideal for complex workflows and reliability. Considering these factors, would be appropriate to justify the protocol choice and ensure optimal performance and scalability for monitoring a large herd of goats.

Response 6: Between lines 317 and 328, the choice of MQTT was better justified. It was added why this protocol is used in IoT applications, its efficiency for asynchronous applications, consumption For this project there was no in-depth study of which protocol would be used, however the suggestion of new protocols is valid and will certainly be considered in future studies.

Comments 7: A block diagram explaining the implementation of your device's firmware is very important to show the reader.

Response 7: A block diagram illustrating the operation of the firmware has been added in the appendix, and it is referenced to the reader in line 246. This diagram was not included directly in the body of the article due to its size and level of detail.

Comments 8: In Section 4.2, it would be important to discuss the coverage achieved with LoRa and provide specific details that are currently missing. Here are some points to consider and clarify for the reader: (1) Frequency and Modulation: Specify that you are using the 433 MHz frequency band for LoRa communication. Also, clarify if you are using the default spreading factor (SF) or if you have adjusted it, as this can impact data rate and coverage. (2) Data Rate: Mention the data rate you are achieving with your LoRa setup. This is crucial for understanding the throughput capabilities of your system. (3) Coverage Testing: Describe any tests conducted to evaluate the coverage of your LoRa system. Mention the environment where these tests were performed (e.g., open field, terrain with hills, forested areas) and discuss the maximum range achieved in these tests. This information helps validate the claimed coverage of up to 15 km from the datasheet in real-world conditions. (4) Comparison and Environment: Compare the theoretical range from the datasheet with your practical results. If the maximum range tested was 720 m (as mentioned in Section 4.5), clarify why this distance was chosen and how it reflects real-world deployment scenarios. (5) Challenges and Considerations: Discuss any challenges encountered in achieving the desired coverage, especially in challenging environments such as areas with hills or dense vegetation as you mentined in the discussion of Table 1. This helps contextualize the effectiveness of your LoRa implementation

Response 8: Points (1) and (2) can be seen in section 5.1 between lines 364 and 368, SF, bandwidth and Coding Rate were added. For comments (3), details were added in section 5.4 regarding where the system is applied, these can be seen between lines 431 and 443. Regarding comment (4), data was introduced in section 6.1 between lines 486 and 495 regarding the use of LoRa. Despite the good results, more studies with this module should be done. Finally, regarding comment (5) about the challenges with LoRA, in line 483 it is said about the limitations that the obstacles generated, these limitations were overcome with the use of SPIFFS and this can be seen in session 7. between lines 616 and 625. The use of more gateways in the same session between lines 626 and 630 is also reinforced.

Comments 9: In Section 5.1, where coverage is briefly discussed, expand on these points to provide a more comprehensive understanding of your system's performance and the factors influencing coverage.

Response 9: Once done, comments were added about the limits of the application case and scope of LoRa, this can be seen between lines 476 and 481 & 486 and 495.

Comments 10: In Section 4.3, it would indeed be beneficial to provide a detailed overview of your data visualization system, including numerical panels and possibly a visual representation similar to Figure 7. Here are some suggestions on how to address this and clarify your system's functionality: (1) Data Visualization System: Describe the layout and functionality of your data visualization system. Explain how numerical panels are utilized to display key metrics or real-time data from the herd monitoring devices. (2) Integration with Telegram: Elaborate on how your application interfaces with Telegram for messaging. Describe the purpose and use cases for Telegram integration in monitoring a large herd. For example, explain how alerts or status updates are sent to Telegram and how stakeholders can interact or receive notifications via this platform. (3)   Real-Time Data Display: Clarify how your system handles real-time data collection and display. Discuss the frequency of data updates and the types of information presented (e.g., GPS coordinates, temperature). Is there a option to export txt data?

Response 10: (1) Figure 4. was added with the description of each panel so that the reader can better interpret what is presented in Grafana. (2) to present to the reader how Telegram was programmed, the lines between 418 and 423 present the connection between it and Node-RED. (3) still in section 5.2 it is presented how the system presents information in real time between lines 407 and 412.

Comments 11: Section 4.4, what is the mass of the system that the goat has to carry around its neck?

Response 11: The mass is 550g, this information can be found on line 250.

Comments 12: In section 5.1 you have an explanation between lines 355-357 about times that I wasn't able to follow, I recommend clarifying. Why do you need 20s to transmit your data, that's not clear to me? Wouldn't it be better to store larger amounts of data to increase the sleep period?

Response 12:  It was better explained in section 6.1 between lines 476 and 481, this time is the device's data acquisition interval. As the device was programmed to remain in deep sleep mode for only 10 seconds to test its performance (described in line 239), it was calculated how long it would take to acquire, send data and wait for the confirmation message. This acquisition interval was 30 seconds for area 2 and 72 seconds for area 1. If the 10 seconds of deep sleep mode are disregarded, the values ​​drop to 20 and 62 seconds.

Comments 13: In Section 5.2, it would be valuable to discuss the issue of nighttime operation and consider implementing a strategy where the equipment enters a prolonged sleep mode during periods of inactivity, particularly when the herd of goats is resting.

Response 13: It was added in session 6.2 between lines 531 and 536. Comments on adapting the device to each grazing pattern were added.

Comments 14: Section 5.3 needs to be substantially improved, there is no information there that demonstrates accuracy or any other feature.

Response 14: This session (now 6.3) has been completely updated, now more GNSS data has been presented. This can be seen between lines 538 and 559.

Comments 15: Conclusion: Recapitulate key findings and contributions of your research. Summarize the performance metrics evaluated and their implications for practical applications.

Response 15: The entire conclusion has been rewritten, now the main data is presented to the reader. This can be seen in session 8 between lines 664 and 688.

I hope that the revisions meet your expectations and that the revised manuscript now complies with MDPI standards. I remain at your disposal for any further clarifications or necessary adjustments.
Thank you again for the time and effort you put into revising this work.

Sincerely,
Mateus Costa

Reviewer 3 Report

Comments and Suggestions for Authors

See attached file.

Author Response

Dear reviewer, I would like to sincerely thank you for the valuable comments and suggestions you provided when reviewing my manuscript. Your comments were extremely useful and helped to improve the quality of the work.

All the yellow comments in the article are new changes. In response to the points raised, I present below a detailed description of the changes made and the necessary justifications:

Comments 1: References. Some of them are too old (i.e., 1, 4, 5, 8, 9, 10, 13, 17, 18, 24, 25, 33, 55,
59, 60, 62, 67, and 72). Please, consider substituting them with similar contributions
published from 2018 on, or alternatively provide reasons to keep them.

Response 1: All cited citations have been replaced by works after 2018.

Comments 2: Abstract. Please, give some hints about the most significant obtained results.

Response 2: The abstract was rewritten, now presenting relevant data from the work.

Comments 3: Section 1. Please, better highlight the paper contributions

Response 3: The references used were about applications and studies in the area, some better study the silvopastoral system and others present their relationship with technologies explored in this article.

Comments 4: Related Works Section. In order to provide readers with a broader perspective
about the topic, I suggest to include the following references [1, 2, 3, 4, 5], but I also
strongly invite the Authors to perform additional research.

Response 4: Thank you for the suggestions, they are great work. I added reference [1] in session 7 on line 642.

Comments 5: Which were the LoRa transmission parameters (i.e., transmitter power output, CR,
SF, BW, etc.) exploited during the tests? How many packets were transmitted?

Response 5:  Isso foi adicionado na seção 5.1 entre as linhas 364 e 368, SF, largura de banda e taxa de codificação foram discutidos nessas linhas.

Comments 6: How many goats were simultaneously involved during the experiments?

Response 6: Somente uma, isso pode ser encontra na sessão 6. na linha 669.

Comments 7: Section 7 . Please, resume the most significant quantitative obtained results.

Response 7: The entire session was rewritten, now presenting the main results of the work, this can be seen in session 8. between lines 664 and 688. Device lifetime data, LoRa range, GNSS accuracy, device rotation on the animal's neck are some of the results presented in this session.

Comments 8: Please, provide hints on future works.

Response 8: During the text, several points were added that should be explored in future work. Between lines 531 and 536, adaptation of the device to grazing patterns is indicated; between lines 654 and 656 further study on architecture and its application in silvopastoral systems is indicated; between lines 623 and 625 it is proposed to study the memory usage on the device; between lines 486 and 491 it is stated that further studies must be done to evaluate the reach of the LoRa module used in this work.

I hope that the revisions meet your expectations and that the revised manuscript now complies with MDPI standards. I remain at your disposal for any further clarifications or necessary adjustments.
Thank you again for the time and effort you put into revising this work.

Sincerely,
Mateus Costa

Round 2

Reviewer 2 Report

Comments and Suggestions for Authors

The paper has improved substantially but I still have the following recommendations:
1) Figure 4 for me is not readable what the authors want to show from (b)-(o)
2) Between lines 199 and 224 there is a description of the sensors that is poor, please create a table and specify the items of your sensors, such as measurement ranges, types of signals and so on, this information is important for the reader and anyone who wants to reproduce the work.

Author Response

Dear Reviewer , thank you for another round of proofreading, in fact these changes are making the article clearer for future readers. 

All the passages underlined in yellow are new changes added to the paper since the last round of corrections.

Regarding your comments, here are the changes:

Comments 1:  Figure 4 for me is not readable what the authors want to show from (b)-(o)

Response 1: Figure 4 shows the layout of the panels created with Grafana to present data to the user. Section 5.2 explains what each panel displays between lines 463-488. This was added as a result of the comment  "In Section 4.3, it would indeed be beneficial to provide a detailed overview of your data visualization system, including numerical panels and possibly a visual representation similar to Figure 7. Here are some suggestions on how to address this and clarify your system's functionality: (1) Data Visualization System: Describe the layout and functionality of your data visualization system. Explain how numerical panels are utilized to display key metrics or real-time data from the herd monitoring devices", available in the previous round. 

Comments 2: Between lines 199 and 224 there is a description of the sensors that is poor, please create a table and specify the items of your sensors, such as measurement ranges, types of signals and so on, this information is important for the reader and anyone who wants to reproduce the work.

Response 2: The entire presentation of the sensors has been improved, and a subsection has been created just for this. In this subsection, the three sensors (GNSS, ADXL345 and SH21) are presented in topics and there is also a table containing their main information (Accuracy, Measurement Range, Resolution and Operating Temperature), this can be seen in subsection 4.1.  With this change, the microcontroller and LoRa have been moved to the "control and communication" subsection 4.2. The device is therefore divided into the "Sensors", "control and communication" and "power system" subsections, available in session 4.

I hope that the revisions meet your expectations. I remain at your disposal for any further clarifications or necessary adjustments.
Thank you again for the time and effort you put into revising this work.

Sincerely,
Mateus Costa

Reviewer 3 Report

Comments and Suggestions for Authors

The paper notably improved after its revision, but some issues are still present. I list them below.

1. Section 1. Please, better highlight the paper contributions. This means that it must clearly stated which are the scientific contributions of the paper. Which are the novelties the paper proposes?

2. As I stated in comment 4 of the last review round, I strongly suggest the Authors to provide readers with a broader perspective about the topic, and I suggest to include all of the references I previously suggested, along with many more the Authors can find by performing a proper literature research.

3. It is not clear what does "Coding Rate 1" stand for (line 365). Moreover, line 367, "KHz" is not correct. Use "kHz" instead.

Author Response

Dear Reviewer , thank you for another round of proofreading, in fact these changes are making the article clearer for future readers. 

All the passages underlined in yellow are new changes added to the paper since the last round of corrections.

Regarding your comments, here are the changes:

Comments 1: Section 1. Please, better highlight the paper contributions. This means that it must clearly stated which are the scientific contributions of the paper. Which are the novelties the paper proposes?

Response 1: highlight that in contrast to the solutions available on the market to replace the shepherd's role by focusing on the real-time location of the flock, the proposal now presented focuses on the seamless recording of the shepherd's work with his flock so that his services can be recognised and fairly remunerated by the administration (in contrast to commercial solutions, the aim is to develop a very low-cost prototype that fully records the programmed locations of the herd without any financial commitment on the part of the shepherd. Only in this way will it be possible to increase shepherds' productivity, income, and thus the ecosystem services they provide. 

to reinforce this point the following changes to the paper were made: 
- Last sentence of the abstract.
- Session 1: between lines 87 - 95.
- Session 7: first paragraph (between lines 667 and 677)
- Session 8: last paragraph (between lines 765 and 769)

These added passages reinforce the highlight of the article.

Comments 2:As I stated in comment 4 of the last review round, I strongly suggest the Authors to provide readers with a broader perspective about the topic, and I suggest to include all of the references I previously suggested, along with many more the Authors can find by performing a proper literature research.

Response 2: To expand more on the related works, all the suggested works have been added, and two more works have also been added. As a result, the "Related Work" section has been changed and this can be seen between lines 152 and 173 and also 191 and 225. The purpose of the related works was not to stray too far from the topic of small ruminant tracking, so not many works involving other animals, such as cows or donkeys, for example, have been added.

Comments 3: It is not clear what does "Coding Rate 1" stand for (line 365). Moreover, line 367, "KHz" is not correct. Use "kHz" instead.

Response 3:  Coding Rate has been changed to Code Rate 4/5 and explained below on lines 438-443, and KHz has also been corrected to kHz (line 442).

I hope that the revisions meet your expectations. I remain at your disposal for any further clarifications or necessary adjustments.
Thank you again for the time and effort you put into revising this work.

Sincerely,
Mateus Costa